# Inertial Measurement of Head Tilt in Rodents: Principles and Applications to Vestibular Research

**DOI:** 10.3390/s21186318

**Published:** 2021-09-21

**Authors:** Romain Fayat, Viviana Delgado Betancourt, Thibault Goyallon, Mathieu Petremann, Pauline Liaudet, Vincent Descossy, Lionel Reveret, Guillaume P. Dugué

**Affiliations:** 1Neurophysiologie des Circuits Cérébraux, Institut de Biologie de l’ENS (IBENS), Ecole Normale Supérieure, UMR CNRS 8197, INSERM U1024, Université PSL, 75005 Paris, France; romain.fayat@ens.psl.eu; 2Laboratoire MAP5, UMR CNRS 8145, Université Paris Descartes, 75006 Paris, France; 3Preclinical Development, Sensorion SA, 34080 Montpellier, France; viviana.delgado-betancourt@sensorion-pharma.com (V.D.B.); mathieu.petremann@sensorion-pharma.com (M.P.); pauline.liaudet@sensorion-pharma.com (P.L.); vincent.descossy@sensorion-pharma.com (V.D.); 4Laboratoire Jean Kuntzmann, Université Grenoble Alpes, UMR CNRS 5224, INRIA, 38330 Montbonnot-Saint-Martin, France; thibault.goyallon@inria.fr (T.G.); lionel.reveret@inria.fr (L.R.)

**Keywords:** inertial sensors, rodents, head tilt, attitude estimation, orientation filters, vestibular system, unilateral vestibular lesion

## Abstract

Inertial sensors are increasingly used in rodent research, in particular for estimating head orientation relative to gravity, or head tilt. Despite this growing interest, the accuracy of tilt estimates computed from rodent head inertial data has never been assessed. Using readily available inertial measurement units mounted onto the head of freely moving rats, we benchmarked a set of tilt estimation methods against concurrent 3D optical motion capture. We show that, while low-pass filtered head acceleration signals only provided reliable tilt estimates in static conditions, sensor calibration combined with an appropriate choice of orientation filter and parameters could yield average tilt estimation errors below 1.5${}^{\circ }$ during movement. We then illustrate an application of inertial head tilt measurements in a preclinical rat model of unilateral vestibular lesion and propose a set of metrics describing the severity of associated postural and motor symptoms and the time course of recovery. We conclude that headborne inertial sensors are an attractive tool for quantitative rodent behavioral analysis in general and for the study of vestibulo-postural functions in particular.

## 1. Introduction

Microelectromechanical inertial sensors are the core components of a multitude of wearable devices used to measure the orientation and kinematics of body parts. These devices have found key applications in healthcare (e.g., for assessing balance, gait and motor deficits, improving motor rehabilitation, monitoring daily activities or investigating work-related physical risk factors) [1,2,3] as well as in sports biomechanics [4]. In animals, wearable inertial sensors have been used predominantly for evaluating the welfare of farm animals and as a quantitative approach for the ethological study of wildlife species [5,6]. Over the past decade, body-worn inertial sensors have also gained momentum as a new strategy for tracking the behavior of laboratory animals [7,8,9,10,11,12,13,14,15,16], offering a cost-effective solution complementary to video-based approaches [17,18,19]. Performing inertial recordings of head movements in rodents is technically simple, as inertial measurement units (IMUs) come in form factors which are small enough to fit on the head of a rat or mouse or to be integrated into existing head implants. As an example, the majority of digital headstage systems used for electrophysiological measurements in rodents now incorporate accelerometers and/or gyroscopes by default. The growing use of inertial data in rodents is however in contrast to the limited perspective on the amount and level of precision of behaviorally-relevant information they can provide.

Measuring head kinematics in rodents is particularly relevant as these animals heavily rely on head movements for exploring their environment. Head mobility contributes to efficient acquisition of olfactory and tactile (whisker-based) information in rats and mice [20,21,22], but is also essential for visual exploration. Indeed, eye movements in rodents occur predominantly in conjunction with head movements [12,15] and seem partially devoted to keeping an overhead binocular field for detecting predators coming from above [23]. Tracking head movements is also a relevant entry point for assessing the ongoing behavioral state, since numerous behavioral activities have a specific head kinematic signature (e.g., grooming, eating, scratching, sniffing, rearing or locomotion) [7,9,21,24].

Head orientation relative to gravity (i.e., head attitude, here referred to as head tilt) is among the key parameters which can be computed using 6-axis IMUs (i.e., containing a 3-axis accelerometer and 3-axis gyroscope). As in humans, abnormal head tilt in rodents is most often associated with a central or peripheral vestibular dysfunction. Experimentally-induced lesions of the vestibular system in rats and mice are a classical approach for studying the associated neuronal compensatory mechanisms [25] and provide useful preclinical models for developing new vestibular rehabilitation strategies [26,27,28]. In such models, precisely estimating head tilt is essential for assessing the severity and recovery time course of postural deficits and, thus, for quantifying the efficacy of candidate treatments.

In the context of experimentally-induced vestibular lesions in rodents, head tilt has traditionally been rated on a discrete scale by an observer, offering a qualitative estimation rather than a quantitative measurement [26,29,30,31,32,33,34,35]. In non-vestibular rodent research, approximations of head tilt angles have occasionally been calculated from photographs or single video recordings [36,37]. Recent efforts in 3D videography have yielded methods for estimating head tilt in rodents [19,38,39], but their accuracy has not been assessed. Finally, head tilt estimates have also been computed using IMU data collected from behaving rodents, either by simply extracting the low frequency component of the acceleration signal [9,11,12] or by employing IMU filter algorithms [8,10,13]. Again, the accuracy of the resulting head tilt estimates has not been properly quantified.

Here, we address the question of the appropriate methodology to obtain reliable head tilt estimates in freely moving rats using readily available IMUs. We first discuss the problem of dealing with sensor offsets and then assess the level of precision of different IMU-based head tilt estimation methods using concurrent 3D video motion capture. Our results show that popular IMU filters offer satisfactory results when implemented with the right set of parameters in conjunction with sensor calibration, with average tilt estimation errors below 1.5° during movement and a precision equivalent to the one of optical motion capture in static conditions. We then illustrate the use of IMUs for quantifying the deficits induced by a unilateral vestibular lesion in rats and propose a series of metrics describing the severity of the associated motor and postural symptoms. Finally, we discuss how IMUs can enhance the sensitivity of preclinical testing pipelines in rodent models of vestibulo-postural dysfunction as well as their general usefulness in the study of rodent behavior.

## 2. Materials and Methods

### 2.1. Animals

Animals were adult Long Evans rats (a 9-week-old 300 g male in the IMU vs. motion capture experiment and 58 12-week-old 200–250 g females in the unilateral vestibular lesion experiment). These animals were housed in standard conditions (12 h non-inverted day/night cycle, 21–24 °C, 50–60% humidity) with free access to food and water as well as the appropriate cage enrichment (cardboard tunnels, shredded paper and dental cotton rolls).

### 2.2. Hardware

Inertial measurements were performed with wired and wireless IMUs (MPTek, Revel, France). Wired IMUs consisted of a small printed circuit board (5 × 7.5 mm) containing a 9-axis digital sensor (MPU9250, TDK Invensense, Tokyo, Japan) and connected to a USB acquisition board via a thin cable. They were used for assessing the accuracy of our offset calibration method (Section 3.1) and for comparing the performance of IMU orientation filters against 3D optical motion capture data (Section 3.2). Wireless IMUs were built using a previously described Bluetooth-based architecture [9]. They consisted of a stack of printed circuit boards containing a 9-axis digital sensor (MPU9150, TDK Invensense, Tokyo, Japan) and a rechargeable coin battery (CP1254, Varta, Ellwangen, Germany). They were used for inertial recordings in the unilateral vestibular lesion model (Section 3.3 and Section 3.4). In the current study, only 6-axis inertial data (3-axis acceleration and 3-axis angular velocity) were used (magnetometer data were not exploited).

### 2.3. Cranial Implants

The most straightforward way of obtaining accurate inertial recordings of head movements in rodents is to attach IMUs onto the skull. For this purpose, a cranial implant was permanently fixed onto the parietal and interparietal skull plates of the animals using dental cement, as detailed in previously described surgical protocols [9,40]. Rats were allowed to recover for at least one week after this procedure. In the case of inertial measurements performed in conjunction with 3D video motion capture (Section 3.1 and Section 3.2), the implant contained two threaded standoffs used to fasten a 3D-printed support containing infrared reflective markers and a wired IMU. For the unilateral vestibular lesion experiment (Section 3.3 and Section 3.4), the implant contained two inverted 3 mm bolts enabling either the fixation of a wireless IMU or the immobilization of the head for video-oculographic measurements.

### 2.4. Unilateral Vestibular Lesion Model

One week after cranial implantation, animals underwent three consecutive habituation sessions (of 5, 10 and 15 min, respectively) to wireless IMU fixation followed by exposure to the behavioral box in the dark (42 × 42 cm with 31 cm walls). Unilateral vestibular lesions were then induced under isoflurane anesthesia by a single trans-tympanic injection of 50 μL of a solution containing either kainate (45 mM, pH 9.5) or arsanilate (184 mM, pH 7.0) dissolved in the same vehicle solution (physiological serum containing the round-window permeabilizer benzyl alcohol at 4% concentration [41]). Injections were always performed in the left middle ear. Animals were allocated to five different groups as follows: the first group received a kainate injection (Kainate group, n = 20) while the corresponding control group received an injection of the vehicle solution (ShamKainate group, n = 11); the third group received an arsanilate injection (Arsanilate group, n = 11) while the corresponding control group received an injection of the vehicle solution (ShamArsanilate group, n = 5); finally, the fifth group (Healthy group) only underwent isoflurane anesthesia without injection (n = 11). Head movements were recorded in the behavioral box in the dark during 20 min sessions distributed as follows: a first baseline session was recorded 1 h before the lesion and, then, a series of sessions were conducted at various time points after the lesion (1 h and then 1, 2, 7, 14 and 28 d). Before each session, IMU offsets were recorded using a three-point tumble test (IMU sequentially placed in three different orientations on a flat surface). Sham animals (ShamKainate and ShamArsanilate) were injected and recorded the same days as their associated lesioned animals (Kainate and Arsanilate, respectively). Because the entire protocol was conducted at different times for the two types of lesions (Kainate/ShamKainate on the one hand and Arsanilate/ShamArsanilate on the other), the two sham groups were not pooled for analysis. Instead, each sham group served as a control for their associated lesioned group.

### 2.5. Sensor Offset Calibration

To assess the level of precision achieved by a multi-point tumble calibration procedure, wired IMUs were fixed onto a custom two-axis motorized rotation stage assembled using off-the-shelf servomotors (AX12A, Robotis, Toulouse, France) controlled by an Arduino Uno. To ensure an homogeneous sampling of tilt orientations, rotation angles were programmed to align gravity with the points of a spherical Fibonacci lattice in the IMU reference frame (Section 3.1 and Section A.1). Each orientation was held for 1.2 s before initiating a new rotation. At the onset of each new rotation, the Arduino was programmed to output a digital signal which was registered by the IMU acquisition board. The 400 ms of immobility preceding the occurrence of a digital signal were used to compute sensor offsets. Gyroscope offsets were calculated as median gyroscope values during these immobility periods. To estimate accelerometer offsets, the average accelerometer values for each axis ($a_{p}$) were first calculated for each immobility period (*p*); offsets ($o_{\mathrm{acc}}$) were then obtained through numerical optimization using the SciPy optimize [42] package by minimizing the average squared difference between the norm of offset-corrected values and 1 g, i.e., by finding:$$\underset{o_{\mathrm{acc}}}{\mathrm{argmin}}\frac{1}{n_{positions}}\sum_{p=1}^{n_{positions}}{\left(1-∥a_{p}-o_{\mathrm{acc}}∥\right)}^{2}$$

### 2.6. Benchmarking IMU Filter Algorithms against Optical Motion Capture Data

#### 2.6.1. Experimental Setup

A rat implanted with a rigid 3D-printed support containing a wired IMU and a set of four infrared reflective markers (3 mm hemispherical markers, MoCap Solutions, Huntington Beach, CA, USA; Section 3.2 and Figure A2) was left free to explore a 1.2 × 0.8 m elevated area around which four infrared cameras (Arqus M12, Qualisys, Gothenburg, Sweden) were positioned. The optical motion capture system was geometrically calibrated to achieve submillimeter accuracy (assessed by computing inter-marker distances). The video and IMU acquisition frame rates were both set to 300 Hz. The camera sync unit was configured to output a TTL signal every 6 frames (i.e., at 50 Hz). This signal was registered by the IMU acquisition board, such that one out of six IMU samples contained a video timestamp. A very small fraction (0.07%) of the intervals between two successive timestamps contained five or seven IMU samples, indicating minor phase shifts between the clocks pacing video and IMU acquisition, respectively. These cases were dealt with as explained in Section A.2 and Figure A1, such that the lag between motion capture and IMU data never exceeded one clock cycle (1/300th of a second).

#### 2.6.2. Initial Calibrations

Prior to the recording session, both systems were calibrated: to compute sensor offsets in the case of the IMU (see Section 2.5) and to establish a global optical reference frame $\left(m_{x},m_{y},m_{z}\right)$ in the case of the optical motion capture system. Intrinsic optical calibration parameters were derived from the properties of the cameras while extrinsic parameters were obtained during a dedicated calibration trial using a hand-held rigid wand and a horizontally-positioned L-frame (indicating the origin of the reference frame and enabling the alignment of $m_{z}$ with gravity).

#### 2.6.3. Motion Capture Data Pre-Processing

The 3D coordinates of reflective markers in $\left(m_{x},m_{y},m_{z}\right)$ were calculated offline using the rigid body tracking functions available in the Qualisys Track Manager software (Qualisys, Sweden). Occasional gaps (i.e., missing marker coordinates for one or two frames) were filled using quadratic interpolation and data were edited as explained in Section A.2 to guarantee sub-clock cycle synchronization with IMU data. Inter-marker distances (Figure A2) were calculated for every frame and used to assess data quality. An overall standard deviation of $\sigma_{d_{12}}=0.44\;mm$ (all recordings pooled, i.e., 36 min of recording) was measured for the distance between markers 1 and 2 ($d_{12}$) but aberrant fluctuations of up to 4–5 mm were observed when the system occasionally failed to properly track the markers (as a result of marker occlusions or spurious reflections caused by urine spots). To work with reliable coordinates, the frames corresponding to the 2% highest values of $\left|d_{12}-median\left(d_{12}\right)\right|$ were excluded. The standard deviation calculated from the remaining frames was $\sigma_{d_{12}}=0.21\;mm$. Using $median(d_{12})$ as a proxy for the ground-truth distance between markers 1 and 2, an estimate of the error of optical head tilt measurement was given by a simple geometrical interpretation: $\arctan\left(\sigma_{d_{12}}/median\left(d_{12}\right)\right)=0.38{}^{\circ }$.

#### 2.6.4. Calculating Head Tilt Using Motion Capture Data

The position of the first three markers (Figure A2) in the global optical reference frame $\left(m_{x},m_{y},m_{z}\right)$ was used to calculate the coordinates of an orthogonal head-bound reference frame $\left(h_{x},h_{y},h_{z}\right)$ as follows:$$h_{y}=\frac{v_{21}}{∥v_{21}∥}$$
$$h_{z}=\frac{v_{23}\times h_{y}}{∥v_{23}\times h_{y}∥}$$
$$h_{x}=h_{y}\times h_{z}$$
where $v_{21}$ and $v_{23}$ are the vectors joining markers 2 to 1 and markers 2 to 3, respectively. The best approximation of head tilt using motion capture data was then obtained by calculating the coordinates of $m_{z}$ in $\left(h_{x},h_{y},h_{z}\right)$.

#### 2.6.5. Calculating Head Tilt Using IMU Data

Head attitude (i.e., head tilt) was computed from IMU data using an extended Kalman filter (EKF) and two classical complementary filters (Madgwick and Mahony) [43,44]. These algorithms were implemented using a custom numba wrapper [45] of the ahrs toolbox (http://ahrs.readthedocs.io, accessed on 12 September 2021), enabling just-in-time code compilation. The output of these filters are a unit quaternion for each sample representing the rotation aligning the *z*-axis of the sensor reference frame with the gravitational acceleration vector while preserving sensor azimuth. Time series of the gravitational acceleration vector’s coordinates were then obtained by rotating $z={\left(\begin{array}{ccc} 0 & 0 & 1 \end{array}\right)}^{⊺}$ by the resulting unit quaternions. Because gravitational acceleration tends to concentrate in the low frequency range of the total acceleration signal [9], we compared these algorithms with a normalized low-pass filtered version of acceleration. Prior to normalization (in order to have a norm equal to 1 g), low pass filtering was performed using a second-order forward-backward Butterworth filter.

#### 2.6.6. Computation of IMU-Based Head Tilt Estimation Errors

Before calculating IMU-based tilt estimates and comparing them with motion capture data, an initial step was to correct for the small misalignment between IMU axes and the head-bound reference frame calculated using reflective markers (see Section 2.6.4). This was achieved by computing the rotation minimizing the point-to-point Euclidean distance between trajectories of the normalized low-frequency (<2 Hz) component of the acceleration vector in the IMU reference frame and of the Earth-vertical axis in the head-bound reference frame during periods of immobility, using SciPy’s implementation of the Kabsch algorithm [46]. This rotation was then applied to raw IMU data before calculating IMU-based tilt estimates as explained in Section 2.6.5.

The time-series of unit-vectors representing synchronized IMU (${\left\{G_{\mathrm{IMU},t}\right\}}_{t}$) and motion capture (${\left\{G_{\mathrm{mocap},t}\right\}}_{t}$) gravitational acceleration estimates were compared by calculating their point-to-point angular distance (${\left\{\theta_{t}\right\}}_{t}$):$$\theta_{t}=\arccos\left(G_{\mathrm{IMU},t}\;\cdot \;G_{\mathrm{mocap},t}\right)$$
This angular distance was calculated for each filter and taken as a measure of filter error in the estimation of head tilt.

#### 2.6.7. Computation of Optimal Filter Parameters

A grid-search approach was used over a broad set of parameter values for each filter (Section 3.2 and Figure A4) in order to determine which one was most suited to the specific properties of head movements generated by freely moving rats. The set of parameters minimizing the average angular error of each filter over all recorded sessions (without splitting periods of movement and immobility) were selected and used for further analysis.

### 2.7. Calculation of Head Tilt Maps

To compute head tilt maps, a spherical Fibonacci lattice of 5000 points was generated and used as vertices for Delaunay triangulation. The number of IMU samples during which the estimated gravitational acceleration vector fell within each triangle was computed to obtain a spherical density map. These spherical head tilt maps were converted to 2D maps using the Lambert azimuthal equal-area projection of the cartopy toolbox (https://scitools.org.uk/cartopy, accessed on 12 September 2021).

### 2.8. Computing the Average Head Tilt Point

The mean direction of a set of three-dimensional vectors on the $S^{2}$ sphere cannot be obtained by averaging spherical coordinates of the points (longitudes and latitudes) because of discontinuities in coordinate values at the poles and on both sides of the meridian with longitude $\pm 180{}^{\circ }$. To circumvent this problem, the estimator for the mean direction of a von Mises–Fisher distribution was used as a measurement of the average head tilt point. This distribution can be seen as an analogue to the Normal distribution on the unit sphere [47], with a mean direction μ and a concentration parameter κ, respectively, analogue to the mean and invert variance of a Gaussian distribution. The maximum likelihood estimate of μ for a set of *N* 3-dimensional unit vectors ${\left\{x_{i}\right\}}_{i\in [\;[1,N]\;]}$ is computed as:$$\mu_{\mathrm{MLE}}=\frac{\bar{x}}{∥\bar{x}∥}$$
where $\bar{x}=\frac{1}{N}\sum_{i=1}^{N}x_{i}$. Note that this formula can only be used when the set of vectors has a preferred direction, i.e., when $\sum_{i=1}^{N}x_{i}\neq \vec{0}$.

This formula was used to obtain the average head tilt point from head tilt maps by computing $\bar{x}$ as the average of vectors representing the centroids of the triangulated sphere’s facets, weighted by the number of gravity estimate samples falling in each of them.

### 2.9. Detection of Periods of Immobility

Periods of immobility were defined as time intervals during which the norm of offset-corrected gyroscope data (i.e., angular speed) was lower than 12° s^−1^. This value was selected to match the separation between the peaks corresponding to periods of immobility and activity in the bimodal angular speed distribution (Figure A3). After applying this threshold, the resulting periods separated by less than 0.1 s were merged, discarding short outliers during immobility. Finally, periods of immobility lasting less than 0.5 s were ignored to avoid false positives due to brief incursions below the threshold during movement.

### 2.10. Calculation of a Circling Index

The quaternion aligning the *z*-axis of the sensor reference frame with the gravitational acceleration vector (see Section 2.6.5) was inverted and applied to offset-corrected gyroscope data (i.e., to the angular velocity vector in the sensor reference frame) in order to obtain gyroscope information encoded in a gravity-polarized reference frame. The *z* component of this rotated angular velocity vector represented the head’s azimuthal angular speed. Its average value computed from periods of movement was converted from [° s^−1^] to [circles min^−1^] in order obtain a more interpretable metric of circling behavior.

## 3. Results

### 3.1. Offset Calibration

Sensor offsets are among the main sources of errors in orientation estimation. They correspond to non-zero intercept values in the sensitivity curves of gyroscopes and accelerometers. While gyroscope offsets can be simply measured as angular velocity values obtained during immobility, accelerometer offsets need to be calculated using static tilt orientations. In the classical multi-point tumble test, offsets are calculated by placing each accelerometer axis in the line of gravity (to register 1 g or −1 g), or perpendicular to it (to register 0 g). Because precisely aligning each IMU axis with gravity can be cumbersome, we opted for a different solution in which accelerometer calibration is based on a collection of 50 different static tilt orientations obtained using a two-axis (pitch and roll) motorized rotation stage. Rotations were calculated such that the resulting gravitational acceleration vector orientations were homogeneously distributed in the IMU reference frame (Figure 1 and Section A.1); as a result, all three accelerometer axes were exposed to similar combinations of acceleration values. Accelerometer offsets were then computed by numerical optimization using acceleration values measured for each orientation (Figure 1; see Section 2.5). To assess the influence of the number of orientations *N*, offsets were calculated using either the full set of tilt orientations (N=50) or a subset of them. Figure 1c shows the average absolute residual error in the norm of the acceleration vector after offset correction as a function of *N*. A saturation is observed for N≥5 with a residual error of 0.0070 g, probably originating from other sensor imperfections (e.g., in the scale factor, response linearity and/or axes alignment). Notably, N=3 yielded a residual error of 0.0085 g when all three orientations were close to orthogonal to one another, corresponding to 93.5% of the optimal error reduction (i.e., obtained with N≥5 points). In conclusion, a manual three-point tumble test is largely sufficient for obtaining reliable offset values on the condition that orthogonality is respected; alternatively, a multi-point automated method such as the one employed here provides a hassle-free solution for sensor calibration.

In our conditions, sensor offsets typically lay in the −0.1 to 0.07 g range for accelerometers and in the −22.6 to 10.5° s^−1^ range for gyroscopes (5–95th percentiles, n=9 IMUs with a sensor temperature between 29 and 33 ${}^{\circ }C$). Both were sensitive to sensor temperature (Figure 1d) but did not vary significantly over a year in a temperature-controlled environment (room temperature: 21–24${}^{\circ }C$, Figure 1e).

### 3.2. Accuracy of Head Tilt Estimation in Static vs. Dynamic Conditions

Orientation relative to gravity (i.e., attitude, here referred to as tilt) is classically calculated from 6-axis IMU data using Kalman filters or a family of less computationally demanding algorithms designated as complementary filters. Because their performance likely depends on movement statistics, we wished to compare these different types of filters in our actual recording conditions, i.e., using head inertial data collected from freely moving rats. To assess the level of precision of an extended Kalman filter (EKF) and of two complementary filters (Madgwick and Mahony) [43,44], we used concurrent 3D video motion capture as a ground-truth measurement of head tilt (see Figure 2a and Section 2.6). Figure 2b shows an example trajectory of the actual (motion capture-based) and estimated (IMU-based filter output) gravitational acceleration vector in the head reference frame. Because gravitational acceleration tends to concentrate in the low frequency range of the total acceleration signal [9], we also plotted the trajectory of normalized low-pass filtered (<2Hz) acceleration. Notably, the actual and estimated trajectories were similar while low frequency acceleration significantly deviated from them in a region visited during belly grooming (dashed ellipse in Figure 2b), a behavior associated with rapid head movements. To assess the performance of head tilt estimation in the presence or absence of movements, we split the data in periods of activity (movement, total duration = 25 min) vs. inactivity (immobility, total duration = 11 min) by angular speed thresholding (Figure 2c and Figure A3; see Section 2.9). Head tilt estimation errors were calculated as the point-to-point angular distance between the trajectories of gravitational acceleration vectors, taking motion capture data as a reference (see Section 2.6.6). The parameters of IMU filters were then optimized to minimize this error (see Section 2.6.7).

In the case of the Madgwick filter, correcting for sensor offsets reduced the optimal average head tilt estimation error by 27% and 45% during movement and immobility, respectively (Figure 2d). Our motion capture data confirmed that the optimal cutoff frequency for using low-pass filtered acceleration as a proxy for head tilt was 2 Hz (Figure A4) [9]. During immobility, optimized IMU filters performed as well as optical motion capture (average error $<0.5{}^{\circ }$) but accumulated variable amounts of error during movement (Figure 2d,e and Table 1): normalized low-pass filtered acceleration became the least reliable (average error $>3{}^{\circ }$) while the EKF and complementary filters yielded comparable average errors (in the 1.1–1.6${}^{\circ }$ range). When splitting the data by angular speed values, however, it became apparent that the EKF was more advantageous in the context of high rotational activity while complementary filters behaved similarly (Figure 2f). The main advantages of the Madgwick filter are its low computational cost and the fact that it only requires to set a unique parameter. In conclusion, while using a normalized low-pass filtered version of the accelerometer signal is appropriate for head-tilt estimate in static conditions, the Madgwick filter seems to be the best compromise for angular speed values up to 100–150° s^−1^ and the use of an EKF is advisable for accurate tilt measurements at higher angular speed values.

### 3.3. IMU-Based Measurements of Head Tilt in a Rat Model of Unilateral Vestibular Lesion

Translational rodent models of vestibular pathologies include lesional approaches consisting in damaging the vestibular organs (the biological equivalent of accelerometers and gyroscopes located in the inner ear) or the vestibular nerve [30,31,34]. These experimental paradigms induce partial or complete loss of vestibular functions and may be designed to allow partial or complete functional recovery. The acute unilateral vestibular lesion model, in particular, is used to reproduce in rodents the main deficits induced by a lateral vestibular concussion or vestibular neuritis in humans (chiefly a head tilt toward and a rapid eye beating phase away from the lesion side) and their progressive recovery. Preclinical pipelines have been established to test candidate treatments for accelerating and/or enhancing recovery, but their sensitivity has been hampered by the use of poorly resolutive head tilt measurements [48,49]. We therefore wished to evaluate the advantages of introducing head-mounted IMUs in such pipelines. We chose to use wireless IMUs to enable seamless head movement recordings in a longitudinal study of postural recovery following a pharmacologically-induced unilateral vestibular lesion. Some animals received arsanilate, a compound inducing an irreversible destruction (cell death) of the inner ear sensory epithelium [31,50], while others received kainate, a treatment inducing reversible excitotoxic synaptic damage [29,30]. IMU recordings were performed before and at various time points after the lesion (see Section 2.4).

Head tilt estimation in rodents still largely relies on observational methods [26,29,30,31,32]. Compared to these approaches, IMUs can provide a comprehensive and quantitative description of head tilt both in static and dynamic conditions. To capture the natural dynamics of head tilt during free behavior, we constructed “head tilt maps” by first triangulating the unit sphere in the IMU reference frame and then counting the number of times gravity was aligned within each triangle (Figure 3a and Section 2.7). Such maps provide a snapshot of the various tilt orientations explored by the animal, with certain areas of the sphere associated with specific behaviors such as grooming and rearing [9]. For visualization purposes, spherical head tilt maps were 2D projected (Figure 3b) and binarized. Differences between treatments were clearly visible on these maps: a strong deviation of the gravitational acceleration vector away from the lesion side (indicating a tilt toward the lesion side) with a moderate reduction of head mobility (the area of the sphere visited by the vector) in the arsanilate group, and a moderate deviation coupled to a strong reduction of head mobility in the kainate group (Figure 3c,d). To obtain a simpler metric of head tilt, we fitted a von Mises–Fisher distribution onto each spherical head tilt map computed during immobility and took its mean direction as an estimation of the average tilt point, i.e., the average direction of the gravitational acceleration vector (see Section 2.8). The trajectories of this average tilt point across all available time points show that arsanilate-treated animals indeed developed a tilt toward the lesion side coupled with a slight downward pitch (Figure 3e).

### 3.4. Quantitative Assessment of Lesion-Induced Deficits and Their Recovery Using IMU Data

Having collected IMU data before and at various time points after the unilateral vestibular lesion, we wished to identify a set of features which best described the severity of induced symptoms and the extent and time course of recovery. One first obvious metric was the time spent immobile (i.e., below an angular speed threshold), which peaked 24 h later for the Arsanilate group compared to the Kainate group (Figure 4a). Notably, the Healthy group, which underwent anesthesia without injection, displayed significantly enhanced immobility up to 1 day after the procedure (Table A2), probably indicating that the effects of anesthesia had not completely worn off at 1 h and that the procedure has indirect effects lasting until the next day. For later time points, enhanced immobility could be attributed to the lesion (Table A2) but was only significantly different from control groups for Arsanilate animals (Table A1). A second feature was the percentage of the sphere visited by the gravitational acceleration vector during movement, which can be considered as a measure of head mobility as well as an indicator of changes in the animal’s behavioral repertoire (e.g., loss and recovery over time of the area corresponding to body grooming; Figure 3b,d). The reduction of head mobility peaked later for arsanilate- vs. kainate-treated animals, and was significantly greater for the Kainate group vs. control groups at t=1h (Table A3), indicating a specific acute effect of kainate (Figure 4b). A third feature was our descriptor of the average head tilt point during immobility, obtained by taking the mean direction of a von Mises–Fisher distribution fitted onto immobility head tilt maps (Figure 3e, see Section 2.8). The angular distance of this point to the animal’s sagittal (xz) plane was mostly significantly affected in the Arsanilate group (Table A5) with an effect peaking at around 40${}^{\circ }$ one week after the lesion (Figure 4c). Finally, the projection of the angular velocity vector onto the gravitational acceleration vector provided a way to quantitatively assess circling (see Section 2.10), a cardinal symptom of peripheral vestibular lesions consisting in a rotational behavior toward the lesioned side. Differences between the distributions of angular velocity values along *z* vs. *x* and *y* were indeed more consistent when considering them in a reference frame aligned with gravity (Figure A5), with a bias of values along *z* indicating a dominant azimuthal rotation toward the left. Notably, this feature was only present in the Arsanilate group (Table A7), with a late-onset kinetics different from the one of other features (Figure 4d) and probably linked to the partial recovery of the baseline activity level (Figure 4a). Overall, these consolidated features provide a comprehensive and quantitative snapshot of lesion-induced postural deficits which can be observed at the level of the animal’s head.

Direct comparison of these results with previous UVL studies is not possible as these studies used a global vestibular rating combining a series of discrete scores (obtained by the observational assessment of head tilt and circling but also of other features such as backward locomotion and postural reflexes). Our measurements are however consistent with the typical time course of this global rating in kainate- and arsanilate-induced UVL: an effect peaking in the first few hours after injection followed by an almost complete recovery within days for kainate [26,29,30] and an effect peaking in the first few days followed by partial recovery over the following weeks for arsanilate [31,32].

## 4. Discussion

### 4.1. Accuracy of IMU-Based Head Tilt Estimation

Using optical motion capture as a ground truth measurement of 3D orientation, we benchmarked IMU-based methods for computing head tilt estimates in freely moving rats. Our results show that all methods achieved the same level of precision (<$0.5{}^{\circ }$) during phases of immobility (Figure 2d). In this context, low-pass filtering the IMU acceleration signal is clearly the simplest solution. During movement, however, optical motion capture remained the most accurate approach while the normalized low-frequency acceleration yielded the least precise head tilt estimates. In contrast, optimizing the parameters of IMU filters (Madgwick, Mahony and extended Kalman filters) could hold the average tilt estimation error below $1.5{}^{\circ }$ (Figure 2d) with a 95th percentile below $4{}^{\circ }$ (Table 1), provided that both accelerometer and gyroscope offsets are corrected (which can be achieved in a satisfactory way using a three-point tumble test as shown in Figure 1c). Such values remain more than acceptable given that in many cases inter-animal differences are potentially a greater source of variability. Overall, the EKF outperformed all other filters but was also the most computationally demanding solution. The Madgwick filter behaved like the Mahony filter but only requires to set one parameter and is thus perhaps the wisest choice for the sake of simplicity. For all IMU filters, head tilt estimation errors linearly scaled with angular speed values above 50° s^−1^ (Figure 2f), a property which needs to be taken into account depending on the regime of movements studied: for intermediate angular speed values (up to around 100–150° s^−1^), using a Madgwick filter seems like a most reasonable solution, while the EKF is probably a safer choice for higher angular speed applications.

### 4.2. Inertial vs. Optical Head Tilt Estimation

Unsurprisingly, a state-of-the-art 3D optical motion capture system achieved a higher accuracy than inertial measurements for estimating head tilt, but also at the cost of cumbersome calibration steps, stringent requirements on the experimental conditions (in particular to avoid parasite reflections) and some data pre-processing dedicated to isolating tracking errors. Optical motion capture nevertheless represents a powerful tool for obtaining ground truth estimates of three-dimensional body movements, an approach adopted in the present work but also successfully used for automatically annotating anatomical landmarks on videos of rodents [19,39]. It nonetheless requires to attach reflective markers on the animal, inducing the same amount of animal preparation work than IMUs. Markerless 3D video approaches can circumvent many of the technical hurdles inherent to optical motion capture and enable the measurement of whole-body movements in rodents [18,19] but likely provide a lower level of angular accuracy than inertial sensors for head tilt measurement.

Despite a lower accuracy compared to optical motion capture systems, IMUs have a number of advantages which make them more tractable and versatile, besides of course offering a considerably cheaper solution. They require a minimal calibration procedure compared to a multiple-camera setup, generate small data files compared to markerless 3D video and do not bear the inherent difficulties of video recordings, such as occlusions, parasite reflections and a necessary compromise between resolution and the size of the field of view. Finally, the need for specifically preparing the animal for inertial measurements will often not be an issue as a growing number of headborne devices such as electrophysiology headstages already incorporate an IMU. For all these reasons, an IMU-based head tilt estimation solution is significantly easier to deploy than most optical approaches, with a minimal compromise in terms of precision. A direct comparison would however be interesting to carry out between IMUs and affordable lighthouse-based 3D localization systems used for virtual reality applications [51].

### 4.3. Application of IMUs to Rodent Vestibular Research

Head tilt and circling are among the main behavioral manifestations of an asymmetrical vestibular impairment in rodents together with nystagmus. Head tilt is of primary interest in the context of preclinical research as it mirrors an equivalent symptom in patients with unilateral inner ear injuries. Contrary to eye movements, the assessment of head tilt in rodents has traditionally suffered from a lack of objective and quantitative measurements. In the case of UVL studies, a battery of static and dynamic features is typically rated on a discrete scale by an observer [26,29,30,31,32]. Head tilt in particular is often rated from 0 to 4 by observing the animal over a couple of minutes. The reported metric is a unique rating combining the scores obtained for all features. While this approach provides a global assessment of vestibular deficits, it remains poorly quantitative and prevents the detailed analysis of individual symptoms.

Here, we show that IMUs offer an attractive solution for a quantitative full scale analysis of head kinematics, including both head tilt and circling but also complementary features such as the degree of head mobility and the animal’s overall level of activity (Figure 4). As illustrated in our model of unilateral vestibular lesion, these measurements can offer a powerful platform for enhancing the resolution of preclinical vestibular research pipelines aiming at identifying potential treatments. They can also be viewed as a complementary approach to recently developed quantitative methods for assessing vestibulo-postural deficits in rodents using video recordings or pressure sensors [33,34,35], which provide information on the alteration of body posture and locomotor patterns.

IMU-based head tilt estimation also offers a route toward a more ethological assessment of vestibulo-postural functions in rodents, such as the ones ensuring head stabilization in space [52]. These functions have mostly been studied using passively applied sinusoidal movements but can now be approached in the context of natural self-generated movements. Similarly, the investigation of how the brain transforms vestibular inputs into a representation of head kinematics can now be transposed to fully unrestrained animals [40].

### 4.4. Perspectives: Quantitative Rodent Behavioral Scoring and 3D Orientation Tracking

Beyond vestibular research, headborne IMUs offer new opportunities for quantitative approaches in rodent behavioral studies. A wide range of different behavioral activities indeed have a distinct signature in IMU data, such as locomotion, rearing, grooming or sniffing [7,9,21,24]. The extent to which 6-axis inertial data can support accurate automated behavioral scoring algorithms remains to be determined. Focusing on intersectional patterns of head tilt and gyroscope signals is probably relevant, as non-gravitational acceleration signals seem to be mostly generated by head rotations in rodents [40]. Another direction is the transition to 9-axis sensors (MIMUs, i.e., IMUs containing an additional 3-axis magnetometer), in order to compute the head’s full 3D orientation. These sensors could be used, for example, in studies of how rodents maintain a neuronal representation of their azimuthal direction or in the measurement of optomotor responses. A similar video-based approach as the one employed here would be necessary to quantify the accuracy of MIMU (i.e., AHRS) filters in freely moving animals.

## Figures and Tables

**Figure 1 sensors-21-06318-f001:**
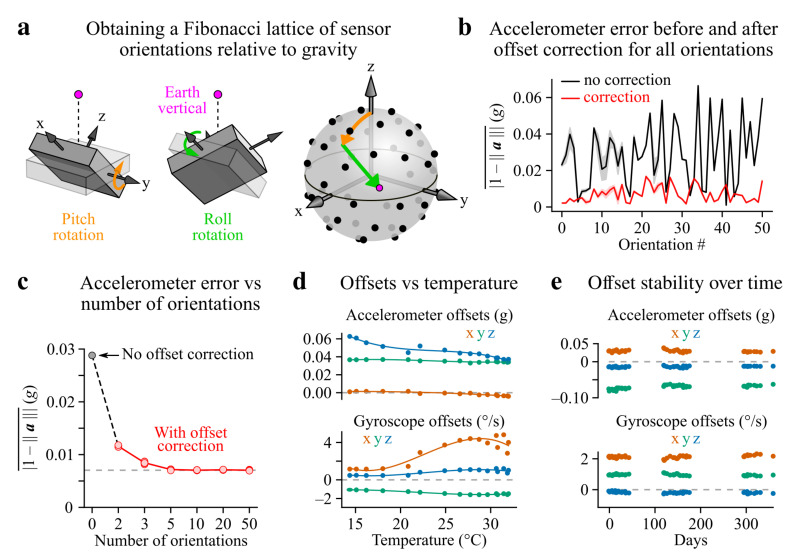
Sensor offset calibration. (**a**) A two-axis (pitch and roll) motorized rotation stage was used to orient the line of gravity (purple dot) along a series of 50 homogeneously distributed directions (black dots) in the sensor’s reference frame (see Section 2.5). The orange and green arrows on the right represent the trajectory of the gravity line during consecutive pitch and roll rotations. (**b**) Absolute acceleration error ($\left|1-∥a∥\right|$, where a is the acceleration vector) during immobility for all tested orientations, with or without offset correction (data from one example sensor, n=10 repetitions of the test). Shaded areas represent the 5–95th percentile range. (**c**) Average absolute error in the norm of measured acceleration as a function of the number of orientations (*N*) used to compute the offsets (data from one example sensor). Red points represent the values obtained with 10 repetitions of the test, with red lines connecting their average values. For N≥5, a subset of the 50 orientations was selected to best match a homogeneous distribution on the sphere for *N* points; for N=2 and N=3, the set of orientations was selected to be the one corresponding to the mode of the error distribution computed using all possible combinations of *N* points, which for N=3 corresponded to orthogonal point configurations. (**d**) Effect of the temperature on acceleration and gyroscope offsets for one example sensor. The lines represent cubic polynomial regressions. (**e**) Stability of gyroscope and accelerometer offsets for one example sensor over a year.

**Figure 2 sensors-21-06318-f002:**
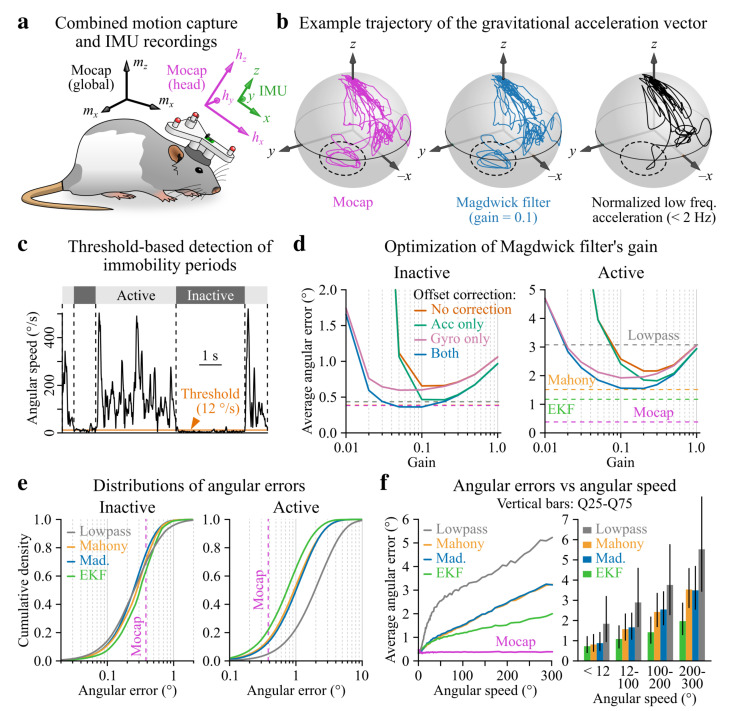
Calculation of the accuracy of IMU-based head tilt estimation using concurrent 3D optical motion capture in freely moving rats. (**a**) Animals were equipped with a head-mounted 3D-printed support containing four IR reflectors for motion capture (here shown in red) and a wired IMU (wire not represented). (**b**) Example trajectories of the actual gravitational acceleration vector (motion capture, left), of its estimate calculated with the Madgwick filter (middle) and of the normalized low frequency acceleration (<2 Hz, right) over a period of 45 s. The dashed ellipse highlights the region visited during a belly grooming episode. (**c**) Example angular speed trace (i.e., norm of the angular velocity vector) and corresponding periods of movement and immobility detected using a threshold of 12° s^−1^ (orange line; see Section 2.9). (**d**) Average tilt estimation error for the Madgwick filter plotted against the filter’s gain, during immobility (inactive, left) vs. movement (active, right) and with or without offset correction. Dashed lines represent the average error calculated with the best set of parameters for other filters as well as the uncertainty in motion capture data (see Section 2.6.3). (**e**) Cumulative distribution of tilt estimation errors for all IMU filters during immobility (inactive, left) vs. movement (active, right). The vertical purple dashed line represents the uncertainty in optical motion capture data. (**f**) Tilt estimation error plotted as a function of the angular speed (left) and median error values and interquartile intervals calculated by bins of angular speed (right). Motion capture error was also plotted to confirm that the uncertainty in the optical estimation of tilt did not increase for specific ranges of angular speed (e.g., due to high intensity movements such as grooming occurring predominantly in conjunction with specific head orientations inducing marker occlusions).

**Figure 3 sensors-21-06318-f003:**
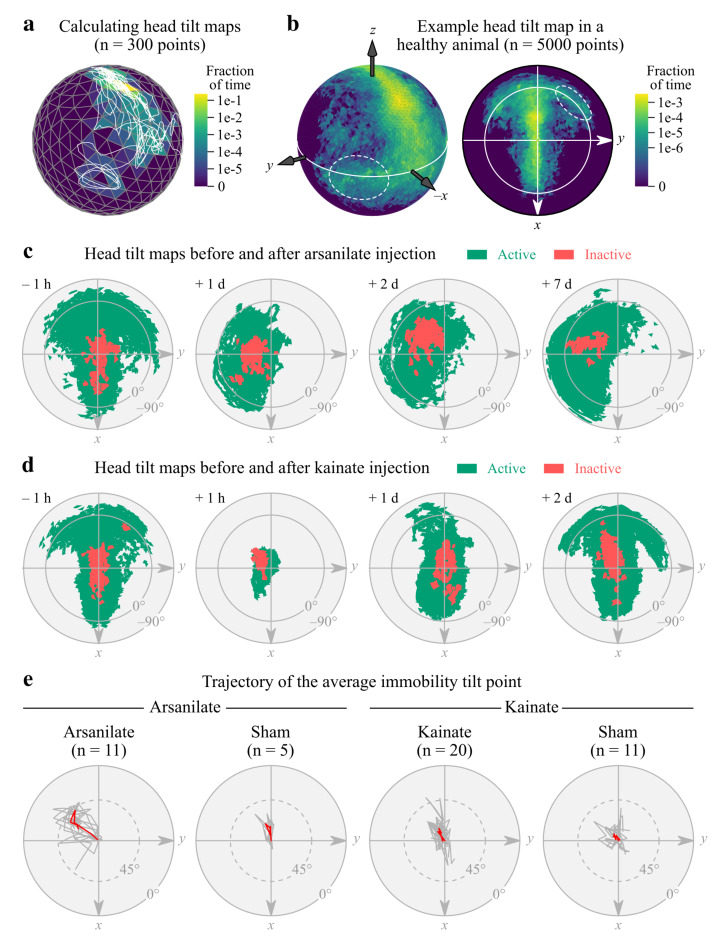
Changes in static and dynamic head tilt following a unilateral vestibular lesion. (**a**) Illustration of the method used to compute head tilt maps: a 45 s trajectory of the gravitational acceleration vector (white line) was plotted in the IMU reference frame on a triangulated unit sphere, with colors reflecting the fraction of time during which the vector passed through each triangle. (**b**) Representative spherical head tilt map of a healthy rat (left) and its 2D projection (right). The white line represents the equator. The dashed white ellipse highlights a region specifically associated with body grooming. (**c**,**d**) Representative binarized 2D head tilt maps before and after a trans-tympanic injection of arsanilate (**c**) or kainate (**d**), computed for active (green) and inactive (red) periods (data from one animal in each case). Note the complete recovery of a normal head tilt map in kainate-treated animals. Injections were performed in the left ear (*y* axis on the maps). (**e**) Trajectories of the average head tilt point during immobility (red curves) for all available time points for arsanilate- and kainate-injected animals and their respective control groups (sham). Individual (per-animal) trajectories and across-animal average trajectories are shown in gray and red, respectively. Individual trajectories were rotated to align the baseline head tilt point (i.e., before injection) with the North pole.

**Figure 4 sensors-21-06318-f004:**
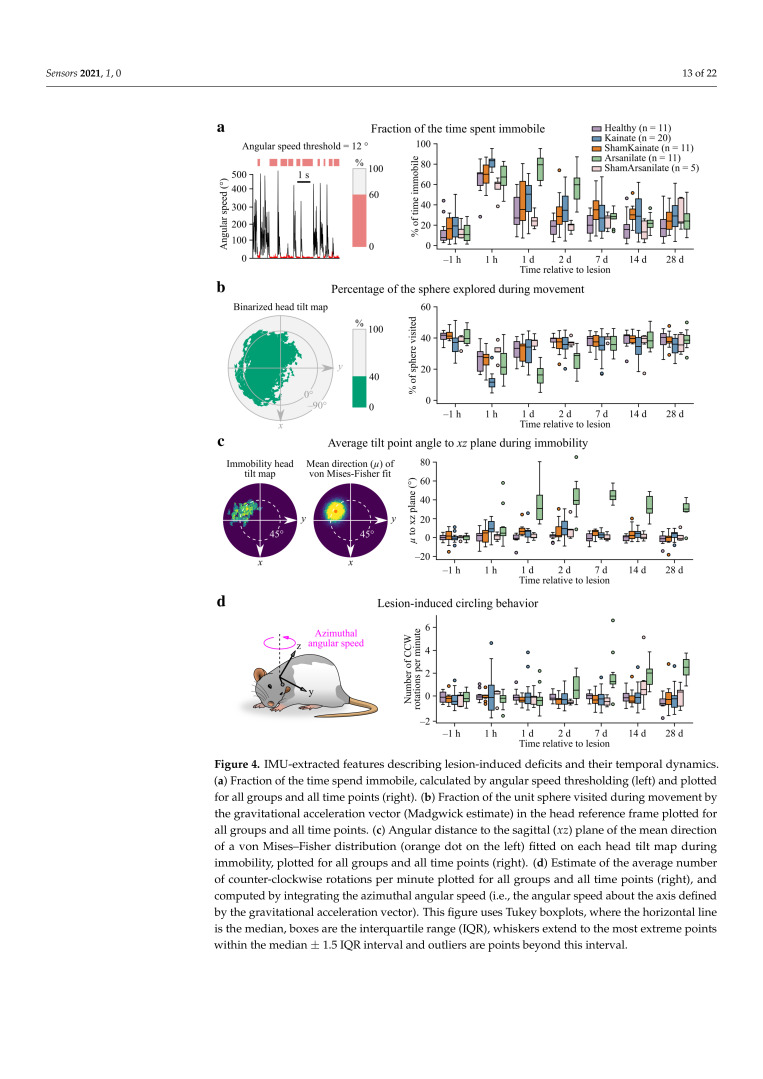
IMU-extracted features describing lesion-induced deficits and their temporal dynamics. (**a**) Fraction of the time spend immobile, calculated by angular speed thresholding (left) and plotted for all groups and all time points (right). (**b**) Fraction of the unit sphere visited during movement by the gravitational acceleration vector (Madgwick estimate) in the head reference frame plotted for all groups and all time points. (**c**) Angular distance to the sagittal (xz) plane of the mean direction of a von Mises–Fisher distribution (orange dot on the left) fitted on each head tilt map during immobility, plotted for all groups and all time points (right). (**d**) Estimate of the average number of counter-clockwise rotations per minute plotted for all groups and all time points (right), and computed by integrating the azimuthal angular speed (i.e., the angular speed about the axis defined by the gravitational acceleration vector). This figure uses Tukey boxplots, where the horizontal line is the median, boxes are the interquartile range (IQR), whiskers extend to the most extreme points within the median ± 1.5 IQR interval and outliers are points beyond this interval.

**Table 1 sensors-21-06318-t001:** Angular errors in IMU-based head tilt estimation for different types of filters. The following optimal parameters were used: cutoff frequency of 2 Hz for the low-pass filter; β=0.1 (filter gain) for the Madgwick filter; $k_{P}=0.3$ (proportional filter gain) and $k_{I}=1.8$ (integral filter gain) for the Mahony filter; $v_{g}=1.0\deg^{2}\;s^{-2}$ and $v_{a}=0.002\;g^{2}$ (noise variance for gyroscopes and accelerometers, respectively) for the extended Kalman filter. Std: standard deviation; Q25, Q75 and Q95: 25th, 75th and 95th percentiles, respectively.

	Angular Error during Immobility (${}^{\circ }$)				Angular Error during Movement (${}^{\circ }$)			
	**Lowpass**	**Madgwick**	**Mahony**	**EKF**	**Lowpass**	**Madgwick**	**Mahony**	**EKF**
Mean	0.43	0.36	0.39	0.44	3.08	1.56	1.52	1.17
Std	0.41	0.25	0.25	0.27	2.58	1.23	1.26	0.99
Median	0.32	0.31	0.35	0.39	2.44	1.27	1.19	0.93
Q25	0.18	0.19	0.20	0.24	1.35	0.73	0.67	0.53
Q75	0.55	0.47	0.52	0.58	4.12	2.08	2.00	1.55
Q95	1.16	0.81	0.83	0.91	7.67	3.83	3.87	2.99

## Data Availability

The analysis pipelines presented in this work can be replicated using code examples and data samples available at github.com/rfayat/sensors_IMU_head_tilt_rodents (accessed on 12 September 2021).

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
