# Peer review of "Inertial Measurement of Head Tilt in Rodents: Principles and Applications to Vestibular Research"

_sensors, 2021, doi:10.3390/s21186318_

Round 1

Reviewer 1 Report

This article is a valuable addition to the vestibular literature and will be of considerable use to future work. The methods are sound, and Dr. Dugué has a history of well-designed rodent behavioral measurement with IMUs. Overall, the paper may benefit from additional proofreading to address minor typographical errors (missing commas, hyphens, and semicolons), to verify sentence structure, and to make sentences more concise, but is well-written and enjoyable to read.

A few minor issues should be addressed:

Methods:

72-73: More information about the one male would be appreciated. Was a larger rat required for motion capture? 

2.4. Unilateral vestibular lesion model: were different vehicles used for kainate and arsanilate? The volumes were equal, and I am assuming that the vehicle contained 4% benzyl alcohol in both groups but am not entirely sure. The ShamKainate group appeared to be more affected by the vehicle treatment than the ShamArsanilate group. I would say that if the vehicle treatment was the same for both groups, they might be considered as a single group; however, if these experiments were carried out at different times and a vehicle group was used in each experiment, these methods are  sound and require minor clarification. Additional clarification to better describe when and how the experiment(s) occurred would be appreciated.

137-138: SciPy reference needed

190-191:  Numba reference needed

208: Citation of Kabsch algorithm needed

280: Gradient descent optimization might be defined prior to results section for clarity.

352-355: Describing the time course of arsanilate and kainate lesions assessed in the cited work might increase the impact of the behavioral data being presented and should be discussed.

353: Please consider adding Vignaux et al. 2012 as a second arsanilate reference -- Evaluation of the chemical model of vestibular lesions induced by arsanilate in rats in Toxicology and Applied Pharmacology.

Table 3: Shamarsanilate should be ShamArsanilate for consistency

Table 7: Shamkainate should be ShamKainate for consistency

Author Response

This article is a valuable addition to the vestibular literature and will be of considerable use to future work. The methods are sound, and Dr. Dugué has a history of well-designed rodent behavioral measurement with IMUs. Overall, the paper may benefit from additional proofreading to address minor typographical errors (missing commas, hyphens, and semicolons), to verify sentence structure, and to make sentences more concise, but is well-written and enjoyable to read.

> We thank the reviewer for his/her comments. We did our best to improve the grammar and typography of the manuscript. Our answers to the reviewer’s questions and suggestions are listed below. Changes in the manuscript are highlighted in blue.

72-73: More information about the one male would be appreciated. Was a larger rat required for motion capture?

> There was no particular reason for using a different rat in the mocap vs IMU experiment (a 9-weeks old male) compared to the ones used in the unilateral vestibular lesion (UVL) experiment (12-weeks-old females). Considering that any adult Long Evans rat was an appropriate subject for the mocap vs IMU experiment, for the sake of reduction (which is encouraged by our local ethics committee), we used an animal which was available in our housing facility rather than ordering an animal matching the gender and weight of the ones used in the UVL experiment. The turn of phrase was indeed a bit misleading (“… except for the IMU vs motion capture experiment…”) and seemed to suggest that this experiment had specific requirements. We changed this for: “Animals were adult Long Evans rats (a 9-weeks-old 300 g male in the IMU vs motion capture experiment and 58 12-weeks-old 200-250 g females in the unilateral vestibular lesion experiment).”

2.4. Unilateral vestibular lesion model: were different vehicles used for kainate and arsanilate? The volumes were equal, and I am assuming that the vehicle contained 4% benzyl alcohol in both groups but am not entirely sure. The ShamKainate group appeared to be more affected by the vehicle treatment than the ShamArsanilate group. I would say that if the vehicle treatment was the same for both groups, they might be considered as a single group; however, if these experiments were carried out at different times and a vehicle group was used in each experiment, these methods are  sound and require minor clarification. Additional clarification to better describe when and how the experiment(s) occurred would be appreciated.

> The vehicle solution was indeed the same for kainate and arsanilate. We changed the text to make this more explicit: “...a solution containing either kainate (45 mM, pH 9.5) or arsanilate (184 mM, pH 7.0) dissolved in the same vehicle solution (physiological serum containing the round-window permeabilizer benzyl alcohol at 4 % concentration).” The ShamKainate and ShamArsanilate groups could potentially be considered as a single group but the two types of lesions were indeed performed at different times (the kainate and arsanilate studies were conducted a few months apart). Each sham group was injected and recorded together with the corresponding test group: in other word, ShamKainate animals were injected and recorded the same days as Kainate animals (same for ShamArsanilate and Arsanilate). As pointed out by the reviewer, this justifies keeping the ShamKainate and ShamArsanilate groups separated. Apparent slight differences between the two sham groups (Fig. 4a,b) are probably due to the fact that the number of animals differ (n = 5 for ShamArsanilate and n = 11 for ShamKainate). We added the number of animals in the boxplot legend in Fig. 4 to make this more apparent. The following text was added to clarify how the protocol was conducted: “Sham animals (ShamKainate and ShamArsanilate) were injected and recorded the same days as their associated lesioned animals (Kainate and Arsanilate, respectively). Because the entire protocol was conducted at different times for the two types of lesions (Kainate/ShamKainate on the one hand and Arsanilate/ShamArsanilate on the other), the two sham groups were not pooled for analysis. Instead, each sham group served as a control for their associated lesioned group.”

137-138: SciPy reference needed

> We added the following reference for SciPy: Virtanen et al, Nat Methods 2020 Mar;17(3):261-272. doi: 10.1038/s41592-019-0686-2.

190-191:  Numba reference needed

> We added the following reference for Numba: Lam et al, LLVM '15: Proceedings of the Second Workshop on the LLVM Compiler Infrastructure in HPC. Doi: 10.1145/2833157.2833162.

208: Citation of Kabsch algorithm needed

> We added the following reference: Kabsch, W. Acta Crystallographica 1976, A32, 922–923. doi:10.1107/S0567739476001873.

280: Gradient descent optimization might be defined prior to results section for clarity.

> Gradient descent is a classical solution for performing numerical optimization. We do not feel that we need to include a detailed description of such a classical methodology, as our code will be made available at https://github.com/rfayat/sensors_IMU_head_tilt_rodents and the SciPy optimize package has all the required detailed documentation. In order not to confuse the reader with unnecessarily detailed terminology, we replace the text in the result section by: “Accelerometer offsets were then computed by numerical optimization using acceleration values measured for each orientation”.

352-355: Describing the time course of arsanilate and kainate lesions assessed in the cited work might increase the impact of the behavioral data being presented and should be discussed.

> We added a last paragraph in the result section 3.4 in which we discuss how our results compare with previous UVL studies. Because these studies only report a global vestibular rating combining the scoring of several features (which is now discussed in the second paragraph of 4.3), we can not directly compare our metrics with the data available in the UVL literature. However, we point out that the time course of our metrics is consistent with the one of this global rating reported in previous kainate and arsanilate studies.

353: Please consider adding Vignaux et al. 2012 as a second arsanilate reference -- Evaluation of the chemical model of vestibular lesions induced by arsanilate in rats in Toxicology and Applied Pharmacology.

> This paper was already cited in the submitted manuscript (ref number 34) but only on one instance. We modified the manuscript to cite this paper in several other relevant parts of the text.

Table 3: Shamarsanilate should be ShamArsanilate for consistency

> Corrected.

Table 7: Shamkainate should be ShamKainate for consistency

> Corrected.

Reviewer 2 Report

The article presents a work aimed at studying the potential of readily available IMUs as instruments from which the head orientation in freely moving rats using can be estimated. The estimation of head movements in rats is the starting point for the definition of metrics that allow quantitatively characterizing the severity of postural and motor symptoms associated with unilateral vestibular lesions in these rodents.

The work is interesting and methodologically correct. In general, the manuscript is well organized, although it would be easier to read if some of the data that appears in the appendices (mainly those related to the results) were incorporated into the body of the manuscript (thus avoiding having to go from body to appendix and come back repeatedly).

Also, the inclusion of a section that presents the state of the art in which the work carried out is framed is also highly recommended. This would allow contextualizing the contribution and real significance of this work.

Author Response

The article presents a work aimed at studying the potential of readily available IMUs as instruments from which the head orientation in freely moving rats using can be estimated. The estimation of head movements in rats is the starting point for the definition of metrics that allow quantitatively characterizing the severity of postural and motor symptoms associated with unilateral vestibular lesions in these rodents.

The work is interesting and methodologically correct. In general, the manuscript is well organized, although it would be easier to read if some of the data that appears in the appendices (mainly those related to the results) were incorporated into the body of the manuscript (thus avoiding having to go from body to appendix and come back repeatedly).

> We thank the reviewer for his/her comments. We did consider including some of the additional figures as panels in the main figures but concluded that it would be counterproductive. Figure A3 is an illustration of the thresholding method used to identify periods of immobility but would have some redundancy with panel 2c and would disrupt the structure of Figure 2. Figure A4 would also unnecessarily complexify Figure 2 while adding few information. Finally, Figure A5 is used to highlight the relevance of computing angular velocity in a gravity-centered reference frame in relation with panel 4d, but again would disrupt the structure of the corresponding main figure. We paid attention to carefully include hyperlinks in the main text when referring to all figures, such that navigating the PDF to consult the appendices should be easy.

Also, the inclusion of a section that presents the state of the art in which the work carried out is framed is also highly recommended. This would allow contextualizing the contribution and real significance of this work.

> We included a new paragraph in the introduction (fourth paragraph) in which we provide an overview of the state of the art in the measurement of head tilt in rodents.